# Digital Islam and Muslim Millennials: How Social Media Influencers Reimagine Religious Authority and Islamic Practices

**Bouziane Zaid** [1], **Jana Fedtke** [2], **Don Donghee Shin** [3], **Abdelmalek El Kadoussi** [4] **and Mohammed Ibahrine** [5,*]

[1]    College of Communication, University of Sharjah, Sharjah P.O. Box 27272, United Arab Emirates; bouzianezaid8@gmail.com
[2]    Liberal Arts, Bard College Berlin, 13156 Berlin, Germany; j.fedtke@berlin.bard.edu
[3]    College of Communication and Media Sciences, Zayed University, Abu Dhabi P.O. Box 19282, United Arab Emirates; dshin1030@gmail.com
[4]    Department of English Studies, Ibn Tofail University, Kenitra 14000, Morocco; malekelkadoussi@gmail.com
[5]    Department of Mass Communication, American University of Sharjah, Sharjah P.O. Box 27272, United Arab Emirates
[*]    Correspondence: mohammad.ibahrine@gmail.com

**Abstract:** Digital platforms have empowered individuals and communities to re-negotiate long-established notions of religion and authority. A new generation of social media influencers has recently emerged in the Muslim world. They are western-educated, unique storytellers, and savvy in digital media production. This raises new questions on the future of Islam in the context of emerging challenges, such as the openness of technology and the often-perceived closedness of religious and cultural systems within Muslim societies. This paper uses a multiple case research design to examine the roles of social media influencers in reimagining Islam and reshaping spiritual beliefs and religious practices among young people in the Gulf Region, the Arab world, and beyond. We used thematic analysis of the Instagram and YouTube content of four social media influencers in the Gulf Region: Salama Mohamed and Khalid Al Ameri from the United Arab Emirates, Ahmad Al-Shugairi from Saudi Arabia, and Omar Farooq from Bahrain. The study found that social media influencers are challenging traditional religious authorities as they reimagine Muslim identities based on a new global lifestyle.

**Keywords:** digital religion; Gulf region; Islam; mediatization; millennial Muslims; social media influencers

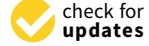



## 1. Introduction

Digital technologies and platforms have provided social media influencers and other users opportunities to produce, consume, and share religious content with various online communities. Social media platforms have become popular among millions of global Muslim millennials residing in the Gulf countries, the Arab world, and beyond (Ibahrine 2014). Muslim millennials are mostly urban, educated, global, and tech-savvy. In the Gulf region in particular, these Global Urban Muslims (GUMmies) are avid users of social media and reside in spaces that afford them modes of consumption, creation and co-creation, participation and articulation outside the constraining limitations of their cultural habitus (Zaid et al. 2021). In the Gulf region and elsewhere in the Arab world, GUMmies have appropriated social media as spaces of interaction, creation, interrogation, contestation of new forms of cultural positioning and of other ways of being and seeing.

Contemporary digital culture has disrupted the traditional practice of religion and the manner in which religion is communicated, consumed, shared, and engaged with. Traditionally, Islamic religious ideas and meanings have been conveyed by established authorities in ritualized forms and in formalized contexts. Religious teachings are a reserved

domain for a small group of professionals that have acquired religious knowledge over the years in traditional Islamic schools (Ibahrine 2014). While they participate in digital culture primarily through YouTube channels, they still sit behind a desk, dress in traditional clothing, wear long beards, speak in classical Arabic, and repeat the same type of knowledge that they themselves acquired through their traditional education and clerical training. The focus of their religious activities is primarily on matters of worship (*ibadat*) rather than civil acts (*muamalat*). Matters of *ibadat* include questions of how to pray, fast in Ramadan, and how to perform the pilgrimage to Mecca.

Hybrid forms of religious teaching that combine education with entertainment have existed in the Muslim world (Wise 2004). Televangelists such as Amr Khaled introduced talk show formats that engage audience participation and provide marketing-inspired testimonials from famous actresses and actors, football players, and ordinary young Muslims (Wise 2004). Some scholars and media analysts suggest that Amr Khaled was inspired by Billy Graham's Christian televangelist performance (Hardaker 2006).

More recently, digital platforms and the rise of social media influencers have empowered individuals and communities to re-negotiate long-established notions of religion and authority. A new generation of social media influencers has emerged. They are western-educated and specialized in non-religious subjects such as communication studies, business administration, computer science, and management. They are fluent in English, hold a cosmopolitan worldview, are good storytellers, and are savvy in digital media production. Their religious practice focuses on storytelling rather than dogmatic texts, on human relationships, civil life, and what it means to be and act as a human being who happens to be a Muslim.

In the context of the rise of social media influencers, some scholars argue that religion shapes and is shaped by consumer culture (Gauthier and Martikainen 2013). GUMmies are urban, highly educated, tech-savvy, hyperdiverse, speak many languages, and are spiritual rather than religious (Sherwood 2016). The mode of consumption that GUMmies emulate does not fit anymore with the traditional religious teachings that consist of teaching dogmatic texts. The predominance and power of consumerism, branding, and promotional culture in the Gulf countries seem to have shaped the way GUMmies interact with religious messages and content when mediated via a new generation of social media influencers.

Campbell (2012) states that "digital religion as a concept acknowledges both how digital technology and culture shape religious practice and beliefs, but also how religion seeks to culture new media contexts with established ways of being and convictions about the nature of reality and the larger world" (Campbell 2012, p. xx). This paper examines the roles of social media influencers in reimagining Islam and reshaping the communication of spiritual beliefs and religious practices among Global Urban Muslims (GUMmies) in the Gulf Region and beyond. Despite the growing importance of social media influencers in effecting the principles and practices of individuals and communities in terms of marketing, advertising, and business contexts, there has been little research on the phenomenon of Muslim social media influencers within the framework of digital culture and digital religion.

After a brief review of the literature on digital culture, social media influencers, social media in the Gulf context, and the theory of mediatization, we describe our methods and analyze the digital presence of four social media influencers in the Gulf Region (Salama Mohamed and Khalid Al Ameri from the United Arab Emirates (UAE), Ahmad Al-Shugairi from Saudi Arabia (KSA), and Omar Farooq from Bahrain) as prominent examples of laypeople reimagining Islam and its current practices.

## 2. Islam and Digital Culture

Kepel (1984), an expert on French Islam, has demonstrated the extent to which religious scholars have used mass media to convey their Islamic messages. He showed, for instance, how Sheikh Abdal-Hamid Kishk used recordings to circulate sermons. Sheikh Kishk popularized the teachings of Islam through his recordings, which have echoed in the population not only in Cairo's streets but throughout the Arab and Islamic world. Along

similar lines, Sreberny-Mohammadi and Mohammadi (1994) argue that it was small media (cassette tapes, photocopies, tape recorders, and telephone usage), not mass media, that played a special role in triggering the Iranian Revolution in 1979 (Sreberny-Mohammadi and Mohammadi 1994).

Research on Islam and the internet emerged in the late 1990s as scholars began to examine the ways in which religious actors and groups used the internet to share religious content and messages (Eickelman and Anderson 1999; Ibahrine 2012). Some scholars suggest that the advent of digital media in the Arab and Islamic World would be in the service of religious leaders to advance their monolithic and orthodox interpretations (Sisler 2006). Others argue that these accounts sound rather unconvincing and simplistic since digital media have a limited effect on culture in the Arab and Islamic World (El-Nawawy and Khamis 2009), and that the digitalization of Islam represents a challenge to the conventional understanding of Muslim identity (Ibahrine 2014; Bunt 2016).

Recent debates in contemporary studies of Islam from a communication perspective address the extent to which traditional patterns of religious life can be sustained in the face of the power of digital media (Bunt 2018). Ibahrine (2018) argues that new media abrades the existing religious hierarchy of the religious establishment, challenges the most glaring monolithic structures and values within traditionally opaque societies, and thus strengthens other religious groups that have been deprived for many decades from obtaining access to the public sphere (Ibahrine 2018). Sorgenfrei (2021) argues that Islam as a social and cultural construct is constantly developing in tandem with societal changes and new technologies. Islam is mediatized in mosques and on mobile devices, and continually changes and adapts to digital culture. The author concludes that even Salafism, a fundamentalist interpretation of Islam, is eager to adopt and adapt to new forms of social mediatization and technological innovation (Sorgenfrei 2021).

Long before the popularity of social media among young Muslims, some jihadist groups realized the power of digital media in indoctrinating and recruiting like-minded potential terrorists (Bunt 2016). The jihadists turned the blogosphere into "Blogistan," to circulate a rigid and linear understanding of the sacred text. El-Nawawy and Khamis (2009) found that online deliberations facilitate neither "rational-critical discourse" along the lines of a Habermasian public sphere, nor *shura* (consultation), *ijtihad* (independent interpretation), or *ijma'* (consensus) in the Islamic tradition of intellectual debate (El-Nawawy and Khamis 2009).

Understanding the relationship between Islam and digital platforms will increase our understanding of the larger cultural shifts at work within traditionally opaque societies. The impact of digital technologies on religious behavior in environments characterized by conservatism and traditionalism can be more profound than in environments characterized by plurality and openness (Ibahrine 2018).

## 3. Social Media Influencers, Islam, and Digital Religion

The subject of religion and social media influencers has only recently begun to attract scholarly attention (O'Brien 2020; Altenhofen 2022; Rozehnal 2022). The focus on social media influencers suggests that the use of social media for religious purposes has moved to another level, characterized by permanency and sophistication. Beyond brands using social media to promote products, social media influencers have also used social media to spread religious and moral messages (O'Brien 2020). Rozehnal (2022) examines new religious expressions of different cyber-Muslims, including religious actors, groups, and communities. These emergent communities are unique in that they display a religious diversity that includes religious clerics, Sufis, feminists, fashionistas, artists and activists, and social media influencers (Rozehnal 2022).

The predominance and power of consumerism, branding, and promotional culture in the Muslim world have shaped the way GUMmies interact with religious messages and content when mediated via social media influencers. Social media has come to encompass

social life at such an unprecedented level and intensity that even the domain of religion can be considered thoroughly digitized (Evolvi 2019).

The literature shows that social media influencers are regarded as influential because of their knowledge and expertise in their respective sectors (Langner et al. 2013). Social media influencers are people who have become celebrities with their followers, although they are "regular" people elsewhere. While traditional celebrities gained recognition because of their professional talent, social media influencers have gained fame by generating a large follower base on social media platforms (Khamis et al. 2017). They are predominantly famous for lifestyle, beauty, fashion, music, and sports. Their influence is measured by the size and demographics of their audience (Ashley and Tuten 2015). They regularly post about their areas of interest on their preferred social media platforms and engage their followers who pay attention to their messages (Abidin 2015). As consumers pay less attention to mass and general advertising, social media influencers have emerged as one of the best marketing communication channels (Carter 2016). The role and functions of influencers in the marketing industry have become so powerful that they changed the whole process and structure of marketing communication.

The rise of promotional culture as a dominant social ethos coincided with the wide adoption of social media influencers and their entry into the religious field. While a large body of literature has already examined social media influencers in different contexts, only a few researchers have analyzed the use of social media influencers for religious purposes (Sorgenfrei 2021). The gap is far greater in the scholarly literature on the practices of Muslim social media influencers. A study found that some Arab countries support Islamic social media influencers to promote a more moderate and tolerant Islam, to push millennials away from listening to extremist religious practitioners, and to combat online hate speech (Farouk and Brown 2021). Unlike conservative religious leaders, social media influencers imbue their religious presence with entertainment content that moves religion from the realm of dogma to the realm of experience and possibility, which appeals more to digital natives in the 21st century (Golan and Martini 2018; Ahmad 2021).

The digitalization and marketization of religious production and distribution have expanded over the past twenty years to create a blended world of commerce and faith where the sacred has become secular and the secular sacred (Ibahrine 2014). Gauthier (2018, p. 388) defines marketisation as "the formatting of given social realities into commodities, or 'goods'". In this specific context, consumerism becomes a means of circulating symbols, and as a result, consumerism is likely to give religion a new shape and new meanings (Gauthier 2018). Digital culture is inherently presentational and promotional (Gauthier 2018), and young people have become socialized into a digital culture that promotes the very notion of individuality, consumerism, branding, and promotional communication. The marketization of religion marks a change in the structures and dynamics of techno-capitalism that celebrates expression of choice, subjectivities, individualities, and identities. This synergy between technology and theology across much of the religious spectrum has become particularly notable in the religious engagements and practices of the generation of young adult Muslims (Trysnes and Synnes 2021).

## 4. Social Media in the Gulf

Much of the existing research on the internet and social media in the Arab World focuses on political activism in the context of the Arab Spring (Ghannam 2011; Abdulla 2013; Zaid 2016, 2018), popular culture and public opinion (Gunter et al. 2016; Khamis 2019), gender and identity (Hurley 2019), gender and empowerment (Zaid et al. 2021), and women's rights (Agarwal et al. 2012; Hafez 2016; Johansson-Nogués 2013; Skalli 2013; Baulch and Pramiyanti 2018; Golnaraghi and Dye 2016). Many of these studies confirm the fact that the affordances of digital platforms seem to have empowered Gulf millennials to challenge established conservative social and cultural practices by providing a space to expand young people's networks, self-present their identities, and question dominant social and cultural stereotypes.

Social media platforms are popular in the Gulf region. YouTube, Instagram, and Facebook are the leading social networking sites, respectively with 31 million, 26 million, and 25 million users (Data Reportal Global 2021). The Gulf region is second to the United States when it comes to the number of daily YouTube views. With 90 million video views per day, Saudi Arabia has the world's highest number of YouTube views per internet user (Data Reportal Saudi Arabia 2021). The popularity of social media platforms in the Arab World has led some scholars to expect that its impact on religious life is likely to rise (Ibahrine 2014; Piela 2022).

A recent study showed that the Gulf has a high penetration rate of social media (Dennis et al. 2018). GUMmies spend an average of two hours a day on social media platforms. Currently, over 92 percent use Facebook, and YouTube is the most popular social media platform in 2021, with 7.89 million users. Almost 79% of the UAE's population has profiles on Facebook, while YouTube's penetration stands at 87.40% (Data Reportal UAE 2021). Mobile social media usage has been booming in the country. The number of users who access social media through their mobile devices is 9.12 million, that is 99% of the UAE population. Instagram and Snapchat have emerged to be the most popular social media platforms among young adults aged 18–24 (Data Reportal UAE 2021). Saudi Arabia's social media usage has rapidly increased. According to Data Reportal Saudi Arabia (2021), Saudi Arabia has a reputation of being the most prominent social media landscape in the Gulf Cooperation Council (GCC) (Data Reportal Saudi Arabia 2021). Its social media influencers have one of the highest local fan bases among GCC countries, ranging from 50 to 60% (Data Reportal Saudi Arabia 2021). Young Saudis believe their Saudi social media influencers to be an authentic voice of their country and themselves (Data Reportal Saudi Arabia 2021). About 84% of the Saudi population live in urban centers with the state-of-the-art infrastructure of internet connections. Social media users stand at 27 million of the total population of 35.08 million (Data Reportal Saudi Arabia 2021). Saudis represent the largest percentage of users on Instagram, Twitter, and Snapchat in the GCC (Data Reportal Saudi Arabia 2021). According to the latest statistics for Bahrain, there were 1.71 million internet users, and internet penetration stood at 99% (Data Reportal Bahrain 2021). About 1.5 million people go online on social media almost every day out of a population of 1.71 million people. The number of social media users is equivalent to 87% of the total population (Data Reportal Bahrain 2021).

The exponential growth of social media has given rise to so-called social media influencers. In the Gulf, the emergence of influencers can be traced to fashion, beauty, entertainment, travel, and lifestyle. Many scholars argue that millennials speak and understand the language of social media. More specifically, recent research asserts that millennials view influencers as people like themselves (Lee et al. 2021). As people spend more time on social media, they will spend at least some time interacting with social media influencers. Millennials in the Gulf countries are more likely than other Arab generations to inform themselves through stories and posts from social media influencers (Dennis et al. 2018). Recent research suggests that female Gulf-Arab social media influencers expand the scope of social and political actors contributing and constructing definitions of reality (Hurley 2019).

## 5. Mediatization and Digital Culture

In her review of the theoretical approaches commonly used in the study of religion and digital media, Campbell (2017) locates three major theoretical frameworks within media studies and religious studies. The approaches include mediatization (Hjarvard 2013), mediation (Hoover 2011; Meyer 2013), and the religious social shaping of technology (Campbell 2017). These approaches have provided important frames for theoretical reflection on the relationship between media technologies and religion, and the ways in which religious discourses and media practices are negotiated in online and offline spheres. This paper adopts mediatization as a theoretical framework to examine how social media as a media institution socializes popular understandings of religion and how this socialization

permeates religious discourses and produces new configurations of religious communities and institutions. The paper uses mediatization primarily to highlight and explain the role of social media influencers in the shifting configuration of Islamic discourse within the Gulf countries and beyond, and how digital platforms and the rise of social media influencers have empowered individuals and communities to re-negotiate and challenge the well-established notions of institutional religious authority.

## 6. Methods

This paper uses a qualitative case study approach to analyze the religious-related content of four well-established social media influencers in the Gulf region. The case study method is useful to explore multiple facets of a contemporary phenomenon in a real-life context (Baxter and Jack 2008; Yin 2017). Stake (1995) differentiates between three types of case study research: intrinsic, instrumental, and collective. An intrinsic case study is typically used to investigate a unique phenomenon, and the researcher has to provide arguments about the uniqueness of the case. The instrumental case study selects a particular case to provide a general appreciation of a phenomenon. The collective case study includes the simultaneous examination of multiple cases in an attempt to reach a broad and more in-depth understanding of a particular issue. The paper uses a "collective" multiple case research design to expand the scope of the investigation with the aim of drawing similarities and differences between the cases for a more in-depth understanding of a complex social phenomenon (Yin 2017; Hunziker and Blankenagel 2021). Adopting a multiple case study method provides an analytical frame to consider the rich cultural, political, and religious contexts of these social media influencers using multiple sources of data.

Our case studies focus on four prominent influencers who produce religious and related phenomena in the digital context in the Gulf region. We acknowledge that the case studies are not representative of digital Islam and the emerging religious contents in the Arab or Muslim world, but the cases are of general importance in that they provide valuable insights to the worldviews of young GUMmies. We are also mindful of the fact that there are other important Islamic social media content creators in other parts of the Islamic world that offer different approaches to digital Islam, but addressing them would be beyond the scope of this paper. We focus on the Gulf region because the social media marketing trends have gained significant momentum in this region. In addition, the market size of the social media marketing business is by far the largest in the Arab region (Arab News 2020).

There are many classifications of social media influencers, including the numbers of followers, types of content, and level of influence (Piriyakul and Piriyakul 2021). The selection of the four influencers is based on their total followings across Instagram and YouTube. All four influencers are well-established in the landscape at the time of writing. They have been active for at least 10 years, have over 1 million subscribers on Instagram and YouTube, and they regularly collaborate with well-known brands. The first two influencers are a married couple, Salama Mohamed and Khalid Al Ameri from the UAE. They have 1.4 million and 1.9 million Instagram followers respectively, and they both have 1.4 million subscribers on YouTube. Ahmad Al-Shugairi from Saudi Arabia has 14.4 million Instagram followers and 2.8 million YouTube subscribers. Omar Farooq has 2.4 million Instagram followers and 4.8 million YouTube subscribers.

The data this paper analyzes consists of Instagram, Facebook, and YouTube posts (videos and text). The analysis focuses on the themes the influencers address in their social media communication that reveals religious significance, denotation, and connotations. The contents were thematically analyzed using Braun and Clarke's (2006) steps of thematic analysis which include: become familiar with the data, search for themes, review themes, define themes, and write-up.

## 7. Results

### 7.1. Salama Mohamed and Khalid Al Ameri

Salama Mohamed and Khalid Al Ameri, a couple from the UAE, are both household names in the country, the Gulf Region, and beyond. Salama Mohamed is an entrepreneur and content creator who started her journey by producing beauty-related content, to later on launch a skincare brand. There is no information on her educational background, but she seems to have graduate level university education. She is articulate in both Arabic and English, and she displays a mastery of storytelling, video production, and editing. Her social media career showcases personal motivation. Using her slogan "blessed with vitiligo," Salama has managed to turn her personal skin condition into a professional career, driven by her interest in skincare. Various skin-care companies like Nivea have used her as a social media influencer for their marketing campaigns.

Khalid Al Ameri's life exemplifies the story of an Emirati social media influencer with a multicultural background, as his mother is from the UK and his father is from the UAE. Khalid went to the American International School in Abu Dhabi and graduated from Stanford University in Business Administration. He initially wanted a career in the media industry and pitched his ideas for TV shows to many TV stations. When he was rejected, he decided to launch his own production company. He is a content creator and a storyteller who started his career using Twitter and Snapchat, and later adopted YouTube, Instagram, and Facebook. Most recently, he has used TikTok to create and convey entertainment content.

Based on the analysis of their social media content, the major theme in Salama and Khalid's communication can be described as 'living as a modern Muslim family can be a fun and enjoyable life without contradicting Islam'. They create and post videos showcasing their marriage, their daily interactions, their activities as a couple, and sometimes their struggles to maintain a healthy marriage. References to Islamic teachings and ethics are always embedded in their storytelling.

Their slice-of-life approach to video creation is a part of their creative strategy. Telling authentic stories about everyday life with a personal touch is one of the strengths of their social media presence. For a conservative society such as the UAE, and unlike the traditional perspective on marriage and family life, Salama and Khalid communicate about their marriage in terms of personal choices where two individuals are free to live their lives the way they like without external constraints from society and culture. In one of his quotes Khalid said: "I feel marriage is an individual experience—this is what we go through. If you can relate to it, great; if you cannot relate to it, fine. We keep it personal. We are showcasing our marriage, our thoughts, our interactions" (Al Ameri 2017). He adds: "It's only when a society is open to different abilities that we truly flourish" (Al Ameri 2017).

Traditional teaching about marriage among Muslim practitioners consists of presentations on the dos and don'ts in marriage, rules and codes of conduct based on the Quran and the Sunna (the Prophet's word and acts). The teaching is dry and the examples are drawn from the life of the Prophet and his companions who lived in 7th century Arabia. GUMmies are used to entertainment and storytelling and might not relate to this style of teaching.

Khalid and Salama use social media to share their family's story, including their sons. They tell stories of how they shop and what they eat during the holy month of Ramadan, what they pack when they travel, and what they do with their kids during the weekend. Both Salama and one of their sons have health-related issues. Salama suffers from a skin condition called vitiligo and one of their sons is a child with autism. Salama is open about her skin condition and has created several videos to address this issue. The couple frame the skin condition as an expression of unique beauty. Khalid created a video to respond to the many questions he received about Salama's skin condition. He responded by saying this: "A lot of people send me direct messages of formulas or medicines that Salama can take to get rid of her skin condition. Why do people make the assumption I want her to change? Why do people make the assumption that she wants to change?" (Al Ameri 2018a).

They created a few videos about their son, telling their followers intimate stories about their experiences with their son's autism, his therapy, his education, and his travel experiences. The focus is always on the positive aspects of their experience and how Islam and Islamic ethics inform their comportment and perceptions.

Another theme in their social media content is addressing social ills such as obesity, excessive use of technology, and healthy eating. Khalid shared his story about his physical transformation where he lost 23 kilos (51 pounds). He explained in detail his diet, exercise routines, and his new outlook on life after the experience. With millennials, this was a compelling way to establish credibility with his audience. Another famous example is a video entitled "Shisha Better than Cigarettes?" Shisha is a tobacco product smoked in a water pipe called hookah. It is a socially acceptable form of smoking in Arab countries, adopted by both males and females. Khalid used the provocative title with a question mark to attract audiences to a topic that has divided people's opinions. He addressed the issue from a scientific perspective, staying away from the usual religious arguments about halal (permissible) and haram (non-permissible). He said: "Shisha contains tar, it contains nicotine. It contains chemicals that can cause cancer . . . I have read reports that smoking shisha for one hour is equivalent to anywhere between 40 to 400 cigarettes . . . because shisha is part of our (Arab) culture and our history, it seems to be given a free pass . . . here's the truth, it is bad for you . . . just because it's part of our past, does not mean it should be part of our future" (Al Ameri 2018b).

Peace is another recurring theme in Salama and Khalid's creative content. Their vision is to connect people, countries, and cultures. They celebrate the notions of tolerance and coexistence of faiths and religions. This theme fits well with being an Emirati. The UAE political leadership promotes itself as an example of diversity and coexistence, where about 200 nationalities and many religions coexist. Many videos were devoted to further explore this dimension of UAE politics with trips and visits to several countries in the Global South. The videos explore various cultures where they express their admiration for cultural differences, similarities, and their commitment to dialogue between civilizations.

### 7.2. Ahmad Al-Shugairi

Al Shugairi is a Saudi influencer. Like Al Ameri, he studied in the United States where he received his BA in Management Systems and an MBA at California State University-Long Beach. He started his career as a media producer and presenter in 2002 on the Middle East Broadcasting Center (MBC). He is most known as the host of the TV show Khawatir, an annual show that aired during the holy month of Ramadan from 2005 to 2015. He currently hosts a new program called Qomrah, which airs on MBC during Ramadan. Al Shugairi once said that "Islam is an excellent product that needs better packaging" (Ben-Ari 2014). He directed this quote to Sheikh Qaradawi, one of the most famous scholars in the Muslim world who represents the archetype of the traditional religious leader. This can be understood as a critique of the established authority of traditional leaders. Despite his global fanbase and popularity, he refuses to brand himself as a religious scholar. Instead, he views himself as a content creator interested in matters of religion (Worth 2009).

Al Shugairi presents Islam as an ethical guidebook that helps GUMmies to creatively rethink religious authority in the Islamic context. He focuses on civic conduct and proper behavior that all Muslims should uphold. This "soft" approach to religion appeals to millennials who are more likely to have a co-creative and open mindset toward religion than previous generations. Like other millennials, Muslim millennials tend to reject traditional institutions and specifically organized religion (Dimock 2019).

One dominant theme in Al-Shugairi's work is to showcase manifestations of societal perfection to argue that such perfection should exist in Muslim societies and that the fact that Muslims do not emulate such perfection is due to their non-compliance with Islamic teachings. Al-Shugairi devoted a whole series of shows to the discovery of what he called "Planet Japan". His focus was on the orderly disposition of Japanese society, its cleanliness, and the character of Japanese people (Al-Shugairi 2013). In each episode, he would cover

some aspects of Japanese culture to draw comparisons between Japanese and Muslim societies. The typical conclusion points to how a non-Muslim society abides by and applies Islamic ethical principles better than Muslims, and that the only difference between Japan and other Muslim countries is ethical conduct. Another related theme is to showcase manifestations of societal decay and explain the causes for such decay to argue that Islam was right to vindicate such causes. He devoted a few episodes to Las Vegas to criticize the excesses of American culture and how moderation, being a core virtue in Islamic ethics, could be the cure.

Like other social media influencers, Al-Shugairi draws his credibility from the work he does and from showcasing the strength of his character. His latest book, *40*, sums up his experience of 40 days of complete seclusion in a small island in the Pacific Ocean. During the 40 days, he stayed away from the internet and social media, fasted and meditated every day, read 40 books, and wrote two self-awareness books (Saeed 2019). He is also open about his own vulnerabilities, admitting that he only became a pious Muslim in his 20s and that he struggled with cigarette smoking as a teenager. This is a form of credibility that is more likely to appeal to millennials than a university degree in religious studies from a traditional Islamic university.

*7.3. Omar Farooq*

Omar Farooq is a young Bahraini filmmaker and content creator. He earned a BA in Mass Communication from the University of Bahrain in 2017. Omar Farooq is in his mid-20s and has already become one of the most famous content creators on YouTube in Bahrain and the Arab World (Farooq 2020). Omar Farooq started by creating and posting short five-minute videos and then moved to longer formats. Omar Farooq uses empathy to create positive human connections (Masri 2019). Omar developed a unique style of experimental storytelling. His most popular series of videos, entitled "Omar Yujarib" ("Omar tries"), consists of stories where he showcases various social experiences, professions, and situations. This experimental storytelling format earned him an impressive following from a wide audience. Examples of these videos include "I am a customs officer" (14.8 million views), "I am pregnant" (13 million views), "I am a mother" (12.4 million views), "I am disabled" (9 million views), "I am a supermarket bagger" (10 million views), and "I am a flight attendant" (11 million views). These videos are a deliberate moral commentary on various human experiences and professions in the contemporary culture of the Gulf countries and the Arab World where injustices, stereotypes, and value judgments are rampant. Omar Farooq goes through the experience himself to show his followers how societal stereotypes and value judgments are in direct contradiction with Islamic ethics.

Another dominant theme in his videos is his travels to various parts of the Muslim world where most people would not dare visit. He has, for example, toured the streets and alleys of Karbala and Mosul in Iraq, visited Saeed in Egypt, and traveled to various parts of rural Pakistan. In Karbala, the holy city for Shia Muslims in Iraq, he conducted street interviews with people to get their views on the coexistence between Sunni and Shiite Muslims. The responses were overwhelmingly positive and people were much more open than viewers might otherwise assume. The video "Sunni in Karbala" currently has 4.2 million views and has about 47,000 comments (Farooq 2021b). Analyzing the comments is beyond this analysis and paper; however, the tendency seems to be positive.

In Pakistan, Omar drove hundreds of kilometers on a motorcycle to visit what he called "the strongest woman in Pakistan." Her name is Naseha, a 76-year-old woman who lived by herself in a hut in a rough mountainous area called Wakhan. She lived off the land and the small cattle she owned. At the end of the video, he asked her if she was happy and she said "yes". (Farooq 2021a). He then quoted a saying from the Prophet that calls for people to be grateful and more accepting of their circumstances.

The "Omar tries" videos can provide the viewers with an "unboxing" experience of traditional perceptions of various professions and experiences in the Arab World and Gulf societies. The notion of unboxing refers to "a genre of videos on YouTube where people

quite literally unbox a product to get a feel for it" (Google 2014). Based on this marketing concept, we interpret Omar Farooq's videos as a type of "unboxing" of professions and situations. For example, "Omar tries" videos unbox how it feels to be a supermarket bagger, a flight attendant, or as a Sunni living in a predominantly Shiite society. Viewers immerse themselves in the process of discovering the human experiences and cultural practices embedded in these professions and situations.

Like Salama, Khalid, and Ahmad, Omar establishes his credibility through his ability to tell stories, to create good quality content, and to showcase his personal experiences. His video production abilities that allow him to shoot videos in various remote sites and to tell compelling, relevant, and relatable stories have earned him the attention and wide following he has so far harnessed.

## 8. Discussion

Hervieu-Léger (2000) argues that, while religion relies on the authority of tradition to sustain itself, it remains a dynamic force capable of adjusting to changing times and will continue to retain its creative potential within our modern contemporary society. The use of new media to convey Islamic or other religious teachings is not new given the mediatization processes in contemporary society (Gauthier and Martikainen 2013). Satellite television and YouTube provided spaces for traditional preachers working in the entertainment industry to challenge the "simplistic view that real religiosity is inherently antagonistic to entertainment, art, and leisure" (Moll 2010, para 3). Social media influencers pushed this idea even further by demonstrating that preaching Islam in an entertaining and artistic way not only maintains Islamic teaching within ethical frameworks, but it is also the only way to reach GUMmies whose modes of consumption do not fit with the traditional religious teachings.

Social media influencers function within the logic of the platform ecosystem and its affordances. They harness the multimodal resources of videos, sounds, images, speech, and text, and the appeals and persuasive powers of branding and marketing to engage GUMmies to experience religion within a digital culture. The religious practice of social media influencers focuses on entertaining storytelling rather than lectures based on dogmatic texts. Unlike traditional religious leaders, they focus on human relationships, civil life, and what it means to be and act as a Muslim in the 21st century. The influencers abide by the social marketing principles of soft sales and value-based selling. They do not pitch their religious message too early; they first entertain their followers with a well-articulated story that the followers can relate to, and then they pitch their message which is usually a form of moral lesson.

In this highly competitive information ecosystem where competition is primarily over audience attention (Wu 2017), Islamic content competes with other media, entertainment, leisure activities, and products. In the context of the ever-increasing pull of promotional culture on all aspects of cultural life, social media influencers are obliged to become more customer-oriented and more entertaining to appeal to the need and wants of GUMmies. The influencers harness data analytics to decide strategically about when and how to engage with their followers to increase their value in their community of followers, and to decide how to target their personalized religious content to strengthen their online communities.

Empowered by social media platforms, the influencers are playing an important role in the shifting configuration of Islamic discourse within the Gulf countries and beyond. They seem to have managed to challenge the well-established notions of institutional religious authority. To gain credibility among GUMmies, and to be worthy of their attention, it is not enough to have a religious degree from a well-established religious school such as Azhar in Egypt or Al-Qarawiyyin in Morocco. It is not enough, either, to speak classical Arabic, wear a long beard, and memorize the Quran to impress the Muslim urban millennials. The latter are still interested in matters of worship (*ibadat*) such as how to pray, fast in Ramadan, and how to perform the pilgrimage to Mecca, but they live in an information and media ecosystem that demands engagement, interaction, immediacy, and personalization. Muslim

practitioners who do not manage their personal brand and reputation by showcasing their mastery of the intricacies of this new ecosystem are probably bound to fail. The influencers establish their credibility and authority by telling their own personal stories, by challenging themselves to lose weight or to self-isolate for 40 days, by producing high quality digital content, by personalizing their content, and by showing their human vulnerabilities.

The findings here echo Gauthier's (2012) argument that consumer societies have triggered a "move from a regime of orthodoxy towards a regime of orthopraxy" (Gauthier 2012, p. 107). Gauthier argues that western societies are moving from a regime of orthodoxy where religious conformity is tied to doctrine and creed (orthodoxy) to a regime where religious conformity is to practice and experience (orthopraxy). The new regime of orthopraxy aims to sacralize the individual and to celebrate the self and the authenticity of personal expression rather than to sacralize a religious tradition or institution. This shift from orthodoxy to orthopraxy can be witnessed in the Gulf and other Arab countries. The discourse on Islam tends to be more focused on civic life and on how to be and act as a Muslim in the 21st century rather than on how to interpret a verse from the Quran. For Gulf and other Arab governments, religious practice is a slippery slope and the focus on creed and the fundamental principles of worship can be a fertile ground for extremism and fanaticism. The focus on orthopraxy constitutes a move away from religious dogma and the risks associated with it. Governments in the Arab world seem to find in these social media influencers the perfect messengers of a new, soft Islam. A shift in the configuration of religious authority is happening whereby the *ancien régime* of religious authority has increasingly struggled to retain religious power while ceding substantial control over the societal and cultural spheres to these social media influencers. This transformation can be interpreted as a consequence of the ability of social media influencers to expand their presence in the digital cultural landscape. This can be best understood as a potential sign of profound cultural change.

## 9. Conclusions

The four cases discussed in this paper show that social media influencers can be considered the new religious leaders in the Muslim digital culture. They play an increasingly significant role as sources of religious content and thus serve some of the functions previously fulfilled by traditional leaders. However, social media influencers lead without being leaders explicitly. Unlike traditional leaders, they are innovative in the sense that they manage to make religious content more relatable, relevant, and appealing to young Muslims.

Particularly in the context of the COVID-19 pandemic, when millions of Muslims could not join public rituals and gatherings in mosques, social media influencers have taken on a new dimension by giving GUMmies access to religious content outside of institutional authority. As suggested in the literature, institutional Islam is losing its former authority and social media influencers are taking over some of their traditional roles as providers of a new moral compass (Hjarvard 2013). The practice of Islam on digital platforms seems to be shifting away from religion as an institution to religion as a practice within a digital culture.

The thesis on the digitalization of Islam and the Islamization of digital platforms (Ibahrine 2018) is still valid and relevant, as social media has become deeply embedded in Islamic preaching. Previous research has shown the effectiveness of social media influencers in conveying religious messages in modern formats and styles. They have the potential to trigger attitudinal and behavioral transformations that can cause changes not only in the understanding of Islam but, most importantly, in the practices of Islam. The emergence of the orthopraxy is likely to enhance individuality, authentic selfhood, and personal expressions, rather than a collective tradition. Consumerism, entertainment, and promotional culture are the driving force behind the new regime of orthopraxy.

Further research is needed to assess the impact of social media influencers on actual behavioral outcomes among Muslim social media users. Future research could perform both qualitative and quantitative research to analyze the new roles that religious influ-

encers play in the formation of new forms of digital Islam and how individualized and personalized religious awareness can be formed among GUMmies.

With the increased power of artificial intelligence and automation on digital platforms, the future of Islamic religious content is unpredictable. Artificial intelligence and bots can generate religious content and fatwas that are personalized. In a fully automated age, imbalances in religious authority have emerged and understanding how GUMmies conceptualize individuality and authority is critically important. The questions on the future of digital Islam in the context of the openness of technologies such as artificial intelligence remain pertinent, given the massive adoption of sophisticated automation processes by social media influencers to personalize communication, religious content, and the delivery of Fatwa bots in a closed religious and cultural system within Muslim societies.

**Author Contributions:** Conceptualization, B.Z., J.F., D.D.S., A.E.K. and M.I.; methodology, B.Z., D.D.S. and A.E.K.; validation, B.Z., J.F., D.D.S., M.I. and A.E.K.; formal analysis, M.I., B.Z.; investigation, B.Z., J.F. and M.I.; writing—original draft preparation, M.I., B.Z., J.F.; writing—review and editing, J.F., B.Z. and M.I. All authors have read and agreed to the published version of the manuscript.

**Funding:** The research received no external funding.

**Institutional Review Board Statement:** Not applicable.

**Informed Consent Statement:** Not applicable.

**Data Availability Statement:** Not applicable.

**Conflicts of Interest:** The authors declare no conflict of interest.

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
