# Peer review of "Digital Islam and Muslim Millennials: How Social Media Influencers Reimagine Religious Authority and Islamic Practices"

_religions, doi:10.3390/rel13040335_

Round 1
Reviewer 1 Report
The topic is interesting, an important issue, and timely for discussion. The message this article is sending to the community is detrimental to the efforts of conveying Islam to the younger generations.
However, it is felt that the discussion tends to be one-sided and sounded like an opinion column instead of scholarly material. The article shapes the narration of how Islam is being conveyed through the influencers, and approves the style of deliveries as the only alternative to the traditional method of preaching Islam-sounded like so. This notion prevails throughout the paper.
It would be best to discuss the history of methods of preaching in Islam and this would highlights characteristics of traditional methods versus so call the new. The connotation that Islamic messages are delivered only through formal means and ‘dry’ is not inclusive of the history in this context. For example, entertainment elements are adopted in conveying Islamic messages long before the existence of digital media platforms. Obviously, the channel and method are relevant to the means of the respective era. Abundance of literature available on this. Scholarly discussion on these frameworks of conveying Islam needs to be included in this paper.
This is indeed a case study by method and of course the influences chosen for this study cannot represent all Islamic influencers on digital platforms. There are different type of Islamic influencers (e.g. laymen, students of islam) as well as methods of conveying messages, and the category chosen for this study represent one out of the many. It would be helpful to include the framework for deriving these four cases so that the scope or positioning of this study will be more precise.
To recheck in text citation. Instead of Reference, Bibliography is more suitable as not all listed articles are cited.
Author Response
Dear Reviewer:
Thank you for your kind comments and suggestions.
Reviewer 1:
The topic is interesting, an important issue, and timely for discussion. The message this article is sending to the community is detrimental to the efforts of conveying Islam to the younger generations.
Thank you for your kind comments.
However, it is felt that the discussion tends to be one-sided and sounded like an opinion column instead of scholarly material. The article shapes the narration of how Islam is being conveyed through the influencers, and approves the style of deliveries as the only alternative to the traditional method of preaching Islam-sounded like so. This notion prevails throughout the paper.
We have rephrased and changed the tone of the “Discussion” section of the paper to minimize the impression that we approve of the influencers' styles as the only alternative.
It would be best to discuss the history of methods of preaching in Islam and this would highlights characteristics of traditional methods versus so call the new. The connotation that Islamic messages are delivered only through formal means and ‘dry’ is not inclusive of the history in this context. For example, entertainment elements are adopted in conveying Islamic messages long before the existence of digital media platforms. Obviously, the channel and method are relevant to the means of the respective era. Abundance of literature available on this. Scholarly discussion on these frameworks of conveying Islam needs to be included in this paper.
We have added a paragraph in the introduction to discuss the use of entertainment in religious teachings. We have also added two paragraphs in the section on “Islam and digital culture” about traditional methods of the use of media technologies by Islamic religious leaders’.
This is indeed a case study by method and of course the influences chosen for this study cannot represent all Islamic influencers on digital platforms. There are different type of Islamic influencers (e.g. laymen, students of islam) as well as methods of conveying messages, and the category chosen for this study represent one out of the many. It would be helpful to include the framework for deriving these four cases so that the scope or positioning of this study will be more precise.
Thank you for your comments. We have added a few sentences in the Methods section to further explain the rationale behind our choice of the four influencers and to acknowledge the limitations of our sample.
To recheck in text citation. Instead of Reference, Bibliography is more suitable as not all listed articles are cited.
We have removed all the references that are not cited in the text.
Thank you very much for your kind comment.

Reviewer 2 Report
This is a fascinating exposition of GCC-based social media influencers who engage Islamic themes and the intersection of religion and everyday life in their content creation. The empirical material is presented in a clear and compelling fashion and the author references relevant theoretical literature on digital religion. There are three areas I would suggest asking the author to clarify/address prior to publication:
- The abstract indicates that "The study found that social media influencers are undermining traditional religious authorities..." This is obviously a strong and intriguing claim. However, the article does not actually present any evidence that traditional religious authorities are being undermined -- only that social media influencers addressing Islamic themes constitute a growing and increasingly popular trend (which is a different kind of argument);
- While the methods section discusses different types of case study research (intrinsic, collective, etc.), there is no full explanation of case study selection or rationale. Why has the author chosen to focus on cases exclusively from the GCC region? What is at stake in so doing with respect to the generalizability of findings? How do GCC-based Islamic social media influences compare to their counterparts in other world regions? The Southeast Asian 'hijrah movement' (largely Instagram-based) is a significant contrasting example with its emphasis on piety; see https://themuslim500.com/guest-contributions-2022/hijrah-movement-a-new-wave-of-islamic-piety-in-indonesia/
- While much of relevant literature on Islam and the Internet has been cited here, there are a few important examples of work that explore the question of how information technology has affected religious authority specifically that have not been cited -- more specifically the works of Rob Rozehnal and Peter Mandaville.
- Finally, there is at least one passage of text that appears to present a worryingly uncritical portrayal of the UAE:
"The UAE is an example of diversity and coexistence where about 200 nationalities and many religions coexist."
To make such a statement without adding additional context about the present of strict social hierarchies with respect to the social position, status, and rights of labour migrants is problematic and sounds more like the kind of image the Emirati authorities would like to project.
Author Response
Dear Reviewer:
Thank you for your kind comments and suggestions.
Reviewer 2:
This is a fascinating exposition of GCC-based social media influencers who engage Islamic themes and the intersection of religion and everyday life in their content creation. The empirical material is presented in a clear and compelling fashion and the author references relevant theoretical literature on digital religion. There are three areas I would suggest asking the author to clarify/address prior to publication:
- The abstract indicates that "The study found that social media influencers are undermining traditional religious authorities..." This is obviously a strong and intriguing claim. However, the article does not actually present any evidence that traditional religious authorities are being undermined -- only that social media influencers addressing Islamic themes constitute a growing and increasingly popular trend (which is a different kind of argument);
We agree. We have rephrased that statement to align our paper’s findings with this claim.
- While the methods section discusses different types of case study research (intrinsic, collective, etc.), there is no full explanation of case study selection or rationale. Why has the author chosen to focus on cases exclusively from the GCC region? What is at stake in so doing with respect to the generalizability of findings? How do GCC-based Islamic social media influences compare to their counterparts in other world regions? The Southeast Asian 'hijrah movement' (largely Instagram-based) is a significant contrasting example with its emphasis on piety; see https://themuslim500.com/guest-contributions-2022/hijrah-movement-a-new-wave-of-islamic-piety-in-indonesia/
We have added a few sentences in the Methods section to further explain the rationale behind our choice of the four influencers and to acknowledge the limitations of our sample.
- While much of relevant literature on Islam and the Internet has been cited here, there are a few important examples of work that explore the question of how information technology has affected religious authority specifically that have not been cited -- more specifically the works of Rob Rozehnal and Peter Mandaville.
Thank you for your suggestions. We have added Rozehnal’s (2022) work in the section on “Social media influencers, Islam, and digital religion.'' We found the author’s input relevant to the discussion. We have also considered Mandaville’s (2003). Since it relates more to issues of Muslim diasporas in the West and identity formation, we found it difficult to incorporate the author’s ideas into the paper due to its slightly different focus.
- Finally, there is at least one passage of text that appears to present a worryingly uncritical portrayal of the UAE:
"The UAE is an example of diversity and coexistence where about 200 nationalities and many religions coexist."
To make such a statement without adding additional context about the present of strict social hierarchies with respect to the social position, status, and rights of labour migrants is problematic and sounds more like the kind of image the Emirati authorities would like to project.
We agree. We have rephrased this section.
Thank you for your kind comments and substantial suggestions.

Reviewer 3 Report
This highly original contribution to digital politics and religion should be published as is.
It's a great read!
Author Response
Dear Reviewer:
Reviewer 3:
This highly original contribution to digital politics and religion should be published as is. It's a great read!
Thank you very much for your kind comment.

Round 2
Reviewer 1 Report
-
Author Response
Thank you for your comments. We have revised accordingly.